# Corporate Social Responsibility and Financial Performance: A Quantile Regression Approach

**Nana Liu [1]**, **Chuanzhe Liu [1,\*]**, **Quan Guo [1,2]**, **Bowen Da [1]**, **Linna Guan [1]** and **Huiying Chen [1]**

1   School of Management, China University of Mining and Technology, Xuzhou 221116, China
2   School of Business, Global Institute of Software Technology, Suzhou 215000, China
\*   Correspondence: rdean@cumt.edu.cn

**Abstract:** A prominent claim within the literature is that corporate social responsibility-disclosured firms are fundamentally more resilient to financial shocks, relative to firms that take no corporate social responsibility action. To test this, we examine the impact of corporate social responsibility (CSR) information disclosure on financial constraints (FC). Our sample is composed of A-share publicly listed firms from Shanghai and Shenzhen in China during 2013–2017. We find that CSR disclosure influences negatively financial constraints. The quantile regression results also indicate that the influences would more obvious when a company faces stronger financial constraints. Further, CSR disclosure influences negatively financial constraints in financially opaque firms, and the effect of financial opaque on the relationship strengthens when the company faces great financial constraints. After considering the problems of missing variables and endogenous, changing the level of CSR and FC measurement, using 2SLS and two-step GMM methods, the conclusion is still robust. However, the results should not be generalized, since the sample was based on 434 A-share publicly listed firms for 2013–2017. From the perspective of FC, this study contributes to the literature in the field of CSR and expands the empirical research on the economic consequences of CSR. It also can encourage enterprises to voluntarily disclose social responsibility information and it is of great significance to promote the stable development of the capital market and society.

**Keywords:** social responsibility information disclosure; financial constraints; financial transparency; quantile regression

---

## 1. Introduction

There has been increasing attention given to the consequences of rapid economic development such as social and environmental problems. According to the results of Rankins corporate social responsibility Ratings (RKS) in 2008, 290 publicly Chinese listed companies disclosed corporate social responsibility (CSR) reports, which increased to 371 in 2009 and increased to more than 851 in 2018. The "Environmental, Social and Governance of Chinese Listed Companies (ESG) Blue Book (2018)" published by the Joint Research Group of the Chinese Academy of Social Sciences and the Responsible Cloud Research Institute revealed the fact that among the 1892 A-share main board listed companies, there were 673 social responsibility reports in 2017, accounting for only 35.57%, and 145 reports were below 10 pages. Although there are many companies that respond actively, the firms that disclose social responsibility reports account for only a small proportion of all Chinese companies. In China, disclosure of CSR remains a relatively casual and spontaneous behavior.

Corporate social responsibility is a complex term broadly defined as the active and (sometimes) voluntary contribution of enterprise resources to actions that are aimed at achieving environmental, social and economic improvements [1]. This issue has attracted more attention, as organizations have realized the strategic importance of such activities. However, the relationship between the

socially responsible practices of a corporation and its financial performance has long been debated [2]. A considerable number of studies have investigated the link between CSR and corporate financial performance [3]. However, the literature has yielded a mixed set of results, including positive [3–5], negative [6–8], neutral [9–11], or even complex [12] relationships, and hence there remains no agreement as to whether or not high levels of CSR activity lead to improved corporate financial performance [4,9]. In this study we hope to provide a different perspective to that presented in the current literature with regard to the relationship between CSR and corporate finance performance by adopting a quantile regression approach. This method allows us to analyze the separate responses of finance performance to CSR at different quantiles of financial constraint distribution.

Does the disclosure of CSR reports really provide useful information? Is it a positive attitude to enhance corporate image? Or does it use this power to conceal corporate earnings manipulation? CSR information disclosure is faced with two opposing theories: social political theory and economic disclosure theory. Social political theory believes the company is pressured by external information demanders to disclose passively the information needed by outsiders; Hence, the company must fulfill a "social contract" to avoid threats, such as legal proceedings [13]. The disclosure of CSR information is a form of "defensive disclosure," which proactively reveal information that is beneficial to itself, and may conceal adverse news, such as environmental pollution. The high level of disclosure does not mean good performance for the firm. Disclosure cannot easily trigger a positive reaction in the market. Based on signal transmission theory, the theory of economic disclosure believes that the company actively discloses information to the public to obtain relevant benefits and show it has excellent social performance and is rewarded by society [14]. This "aggressive disclosure" demonstrates the firm's management capabilities and performance. The rise in the level of disclosure scores improves the economic consequences. Therefore, studying the economic consequences of CSR information disclosure helps determine which theory can explain the CSR disclosure behaviors of publicly listed companies.

Besides, CSR report is an important part of non-financial information to public. This information is general not reported in financial reports, however brings about great implications for assessing firm value. Hence, identifying how financial and non-financial disclosures interact with each other to affect corporate financial constraints is an interesting feat. We also have mainly been interested in question identifying whether CSR and financial disclosures substitute or complement each other in affecting the financial constraints.

In addition to the substitute or complement function on affecting the financial constraints, we also provide evidence based on the Chinese capital market. China is one of the emerging markets that reflect the importance of nonfinancial information obviously. Moreover, a number of related study maintains that under financial constraints situation, the form external financing way was selected by more developing financial market than the developed markets [15]. Hence, our review could provide evidence for the influence of CSR on financing and also understand the role of financial transparency between CSR and FC.

Our study covers 434 companies with social responsibility disclosure scores in RKS, based on annual business data from 2013 to 2017. We adopt panel fixed effect model and panel quantile regression method. Our findings show CSR disclosure is negatively linked with financial constraints. The negative link between FC and CSR is stated when the firm faces strong financial constraints. This negative influence is significantly stronger when the company financial opacity is great, thereby indicating a potentially relationship between CSR and FC. These results are robust when we try to control numerous potentially depended variables and substitute some key variables and methods.

Our research contributes to extant study on the economic impact of nonfinancial disclosure in different ways. First, we predict that the impact of CSR on financial constraints is not constant across the cash flows distribution as typically assumed by previous research. We demonstrate that CSR generates distinct consequences on FC at different parts of the cash flow distribution. The results of the quantile regression method indicated that the effect of CSR performance is quantitatively larger for firms facing substantial financial constraints than those facing minimal financial constraints. Our adoption of

quantile regression method makes us to get a more complicated feature of the relationship between CSR disclosure and financial constraints. Second, we examine the interactive relationship between financial and non-financial disclosure. Francis, et al. [16] found that the level of transparency as reflected in the firm's financial reports is negatively linked with the cost of equity capital. Our studies complement theirs by focusing on non-financial disclosure. We suggest that the influences of these two forms of disclosure are probably substitutive in their ability to lessen financial constraints. This result devoted into understanding the interactive features of the two different kinds of disclosure ways.

The rest of our paper is organized as follows. The relevant literature reviews and hypotheses are in Section 2. The methodology and key variables used are presented in Section 3. The empirical and robustness test results are reported in Section 4. In Section 5, we list the conclusions and references on the study. Finally, the Appendix A contains a detailed list of the tables.

## 2. Literature Review and Hypotheses Development

Corporate social responsibility, as both a concept and practice, has become popular in the researches. Backman (1975) [17], Davis (1960) [18] and Manne (1972) [19] defined the meaning of CSR by considering social responsibility as the opposite of economic responsibility. McGuiire (1963) [20] divided the responsibilities of enterprises into economic, social and legal responsibility. Social responsibility mainly means that enterprises should pay attention to politics, social welfare, education, employee benefits and other social interests. Preston (1975) [21] first analyzed CSR based on a process perspective and proposed a "corporate social responsibility matrix". Some scholars began to systematically integrate relevant theoretical results, and gradually formed the concept of "corporate social performance". Carroll (1979) [22] constructed a three-dimensional model of corporate social performance. Wartick and Cochran (1985) [23] inherited Carroll's three-dimensional model. They believe that identifying social issues is important, but more important is how to manage social issues. Wood (1991) [24] mainly revised the principle dimensions of the Carroll model.

There are different opinions about how to measure that performance. The most widely used applications are reputation index method, content analysis method, questionnaire survey method and measurement based on professional institution database. In 1971, the US Economic Priorities Committee (CEP) ranked the 24 companies in the paper industry for pollution control. Vance (1975) [25] used the same method to ask 86 company employees to sort and rank 45 large companies. A large number of empirical studies used the reputation evaluation results of Fortune [26–28]. Content analysis usually refers to the method for quantifying corporate social responsibility information, coding and classifying qualitative information of company reports or document disclosures [29,30]. The questionnaire survey method evaluates CSR based on the scores and dimension scores of each item in the questionnaire [31,32]. The KLD Index Method is a measurement method created by the independent research and rating agency KLD (Kinder, Lydenberg, Domini and Co. Inc.) in the 1990s [33]. The studies exploring the possible links between CSR disclosure and key financial decisions or parameters have substantially increased in the last decade. Recent study works which included investment or future cash flows [34], systematic risk [35], the cost of debt/bank loans [34,36–38] and so on. Cheung [39] examined the indirect relationship between CSR and firm cash flows depends on the market value perspective.

The information disclosure system is the guarantee for the normal operation and healthy development of the capital market [40,41]. Full information disclosure can reduce information asymmetry in the market transaction process [42]. It enables enterprises to gain the trust of investors and the recognition of the public. Richardson, Welker and Hutchinson [40] found that CSR information published by enterprises affects economic decisions through the three theories of information asymmetry: principal-agent theory, stakeholder theory, and social responsibility theory [43–46].

Freeman(1984) [47] proposed that managers should tailor their policies to satisfy numerous constituents, not just shareholders. These stakeholders include workers, customers, suppliers, and community organizations. Clarkson (1995) [48] argues that stakeholders take risks because they invest in corporate capital, and that these individuals or organizations are divided into primary and

secondary stakeholders based on how close they are to the business. Mitchell (1997) [49] considers there are three attributes of stakeholders: conforming to certain regulations, power, and urgency. The concept of principal-agent relationship was first proposed by Ross (1973) [50]. According to Jensen and Meckling (1976) [51], a principal-agent relationship is a contract in which one or more actors employ other actors to provide services and pay compensation according to the quantity and quality they provide. This paper believes that stakeholder theory is one of the important theoretical foundations of CSR, and clarifies the content and scope of relevant subjects of CSR, which is conducive to people's understanding of the concept of CSR.

Regarding the possible links between CSR and economic reward, most of studies found that CSR could significantly relieve corporate FC [52,53]. Preston [54] and Sturdivant and Ginter [55] argued that social responsibility is positively related to financial performance. Firms which earning superior CSR performance are more likely to get lower loan costs than those in inferior CSR performance. Li, et al. [56] and Goss and Roberts [36] argued that companies which have outstanding social responsibility prefer to achieve a lower bank call rate and longer loan term. Chan and Unger [57] have found the firm's CSR is negatively related with FC by using the index and Z-score as the measurement of FC. Platonova et al. (2018) [58] proved that there is a significant positive relationship between CSR disclosure and the financial performance. Bae et al.(2019) [59] found that CSR reduces losses in market share when firms are highly leveraged. By reducing adverse behavior by customers and competitors, CSR helps highly leveraged firms keep customers and guard against rivals' predation. Ok and Kim (2019) [60] found that firms with better corporate social responsibility (CSR) performance generally exhibit cheaper equity financing. Cupertino et al. (2019) [61] found that the environmental and social firm performance positively impacted corporate capital accumulation using a sample of US manufacturing firms from 2002 to 2017.

By contrast, CSR information disclosure may put the company at a disadvantage in the market competition. Anderson and Frankle [30] concluded that excessive CSR information disclosure could also negatively impact the whole capital market. Platonova et al. (2018) [58] use OLS estimator approach to analyze the CSR related data. Their findings indicated a significant negative link between CSR and financial performance of commercial banks in Vietnam. In addition, based on the analysis of information asymmetry theory, an interactive U-type nonlinear relationship may exist between them, and some studies argue that there are no concern about CSR and firm performance [62]. Dai et al. (2019) [63] found that there is an inverted U-shaped nonlinear relationship between CSR information disclosure and stock price crash risk.

We predict that CSR disclosure will help investors fully understand the future value of enterprises and alleviate adverse selection. It may also improve the financial constraints faced by enterprises to a certain extent. In addition, it may encourage the firm to strive for additional financial support for enterprise investment projects to help reduce financing costs and expand financing scale, thereby alleviating financial constraints. Furthermore, this mitigation effect becomes obvious when the company faces strong financial constraints. Our predictions are as followings:

**Hypothesis 1:** *CSR information disclosure influences negatively financial constraints.*

**Hypothesis 2:** *The negative influences is more obvious when company faces stronger financial constraints.*

Enterprises have comprehensive and real CSR information, such as environmental protection measures and amount of environmental protection investment. If these information and figures are not disclosed, then external investors would not know these efforts. Debt financing is the process of obtaining as many loans as possible. These funds will be used to expand a firm's operations, improve performance, and increase profits. As a form of information advantage, a firm may choose to conceal negative news about the company or only disclose favorable information. It may even invest in some environmental damage and non-sustainable development projects. Hence, potential investors, such as creditors, may make wrong investment decisions because they cannot assess the

real operational risks of the company. However, these firms ultimately bear the losses caused by environmental litigation and fines caused by environmental problems. Previous studies have found that information disclosure behavior can greatly reduce information asymmetry and agency problems. High-quality information disclosure which ranked in the top 100 according to the monitoring data of RKS can enhance communication between companies and investors. It can also reduce additional expenses. However, does the influence of CSR disclosure behavior on FC differ when the transparency of financial information varies?

Numerous studies found a negative relationship between financial transparency level and the cost of equity in U.S. [64]. Francis et al. (2005) [16] and Hail and Leuz (2006) [65] extended this research scale to international settings. Dhaliwal et al. (2012) [66] believed that the connection between analyst forecast error and CSR disclosure is not positive in companies and countries owned great financial opacity.

We believe that when the firm's financial transparency is low, the disclosure of high-quality social responsibility reports could convey additional nonfinancial information to the market. It also reduces information asymmetry, thereby decreasing the cost of equity capital and alleviating financial constraints. Thus, the disclosure of social responsibility reports is the "marginal contribution" of reducing information asymmetry is significant. By contrast, the disclosure of high-quality social responsibility reports may not effectively reduce information asymmetry when financial transparency is high. Hence, companies cannot enjoy the benefits of lowering the equity capital cost.

The impact of disclosure motivation cannot be ignored. A single company cannot easily change the governance environment in a perfect and fully competitive market. It can decide if and how to disclose social responsibility reports according to the level of financial transparency because the firm's financial transparency could be controlled through real transactions or accounting treatments.

Impression management theory by Elsbach and Sutton [67] believes that a company with bad financial performance may conduct corporate impression management by disclosing nonfinancial information, such as social responsibility reports [67]. Thus, the possibility that the cost of equity capital will increase due to poor financial performance is reduced. The social responsibility report became a window-dressing, self-interesting tool for opportunism. The company was only called a "good actor." At this point, the firm itself may increase the cost of equity capital due to financial opacity. However, because the disclosure of nonfinancial information such as social responsibility "hedged" the influence of financial opacity, the cost of equity capital will decrease alongside financial constraints. By contrast, the role of social responsibility information disclosure may not be obvious when financial transparency is high. Therefore, we state the following hypotheses:

**Hypothesis 3:** *Financial transparency negative influences the relationship between CSR disclosure and FC.*

**Hypothesis 4:** *The negative effect of financial transparency on the relationship between CSR disclosure and FC strengthens when the company faces stronger financial constraints.*

## 3. Data and Methods

*3.1. Data*

The CSR data are obtained from RKS, based on CSR and annual financial report of companies, which are used to evaluate CSR performance. These data are among the major measurements for a firm's social responsibility in China. We applied the China Stock Market and Accounting Research database and the Wind Financial Database as primary sources, and searched the data mainly from IPO firms in China from 2013 to 2017. This study begins from 2013, because the number of companies which decided to disclosure company information due to the social responsibility in China increased significantly in this year. For the more precise results, we exclude the firms in financial industries and

special treatment (ST). We derive 2170 firm-year observations from 434 firms in 5 years. All of the continuous variables are minorized at the 1% and the 99% levels.

*3.2. Variable Definitions*

　　Corporate social responsibility (CSR): The CSR data are obtained from the RKS, based on CSR and annual financial report of companies, which are used to evaluate CSR performance. These data are among the major measurements for a firm's social responsibility in China. RKS has set up 15 first-level indicators and 63 second-level indicators(strategy, governance, stakeholders, labor, operations, consumers, content balance, information availability, reporting innovation, credibility and transparency, normativeness, availability and effectiveness of information transfer, community involvement and development, mitigation and adaptation to climate change information, social investment information and so on) from four aspects: macrocosm, technical, content and industry. The scoring method is that the weight of M is 30%, out of 30 points; the weight of T is 15%, out of 15 points; the weight of C is 45%, out of 45 points; and the weight of I is 10%, out of 10 points (the comprehensive industry and other manufacturing industries have no the I indicators, so the weight of M is adjusted to 50%, the T weight is adjusted to 20%), the total score of CSR is a summary of the scores of the four part.

　　Financial constraints (FC). Following Kaplan and Zingales [68] and Poncet, et al. [69], we use firm cash flow as the dependent variable. The relationship between free cash flow and financing constraints is the opposite. Thus, enterprises with more cash flows will face lower financing constraints than those with lesser cash flows. In robustness check, the SA index to proxy stands for cash holdings. SA index has two advantages: (1) the calculation method is simple and convenient; (2) the financing constraint index uses only two variables of firm's size and age. Hence, it could effectively avoid endogenous variables to financing constraints. The impact of this measure on corporate financing constraints has been widely used by many scholars, such as Hadlock and Pierce [70]. The formula is as follows:

$$SA = -0.737*SIZE + 0.043*SIZE^2 - 0.04AGE \qquad (1)$$

where SIZE is the logarithm of the company's total assets.

　　The index calculated by Equation (1) is negative. The larger the value of SA, the smaller the financial constraints faced by the enterprise will be, thereby indicating a negative relationship.

　　Financial transparency (ABSEM). Consistent with Dhaliwal, et al. [71], we use the absolute value of accrual earnings management (ABSEM) level obtained by modifying the Jones model to measure financial transparency. The higher degree of ABSEM, the lower the reliability and transparency of the company's financial information will be. The formula for ABSEM is

$$\frac{TAC_{i,t}}{TA_{i,t-1}} = \beta_0 \frac{1}{TA_{i,t-1}} + \beta_1 \frac{\Delta Sales_{i,t} - \Delta AR_{i,t}}{TA_{i,t-1}} + \beta_2 \frac{PPE_{i,t}}{TA_{i,t-1}} + \varepsilon_{i,t} \qquad (2)$$

where i indicates company; t represents year; TAC is total accrued profit; TA denotes firm's total assets; $\Delta$Sales, $\Delta$AR, and PPE represent an increase of main business income, accounts receivable, and fixed assets, respectively.

　　Ownership structure (SOE). SOE is equal to 1 if the ultimate controller is the state and 0 otherwise, according to the CSMAR database.

　　We also include some control variables, including firm size, growth, net income divided by total asset, total liabilities divided by total assets, and total share ratio of the top 10 shareholders and so on [72,73]. The definitions of all variables are presented in Appendix A.

### 3.3. Research Methods

We first use panel fixed effect model to examine the influence of CSR on firm's FC. The formula is as follows:

$$FC_{it} = \alpha_0 + \beta_1 CSR_{it-1} + \beta_2 SIZE_{it} + \beta_3 GROWTH_{it} + \beta_4 ROA_{it} + \beta_6 LEV_{it} + \beta_6 TEN_{it}$$
$$+ \beta_7 SOE_{it} + \mu_i + \lambda_t + G_r + \varepsilon_{it} \tag{3}$$

where i denotes firm, and t represents year; CSR is the reported corporate social responsibility score, and FC is the firm's financial constraints.

We refer the prior studies [74] to use the control variables. Specifically, these control variables are operating income growth rate (GROWTH), return on total assets ratio (ROA), debt–asset ratio (LEV), proportion of the top 10 shareholders (TEN), and state-owned characteristics (SOE), among others. The full definitions of all variables are provided in Appendix A. The lag values of the independent variable are used because the economy of social responsibility disclosure experiences a hysteresis effect. In addition, we account for industry and year fixed effects to control for the effect of time-related industry patterns and macroeconomic uncertainties. A positive (negative) $\beta_1$ in Equation (3) indicates a positive (negative) effect of CSR on a firm's cash flows and an opposite influence on financial constraints.

The panel fixed effect model built on the assumption that the average effect of CSR on FC is constant. CSR disclosures may also affect the dispersion of financial constraints and generate distinct effects on different parts of the firm cash holdings distribution. Thus, the panel fixed effect method used above may conceal significant parameter heterogeneity in the link between social responsibility performance and cash flows. We then apply the quantile regression method (QR) to solve this problem. The formula showed as follows:

$$Q_{y_{it}} = (\tau_k | \alpha_i + x_{it}) = \alpha_i + \beta(\tau_k) x_{it} \tag{4}$$

where i and t stand for firm and year, respectively; $\alpha_i$ epresents the unobservable individual effect; $\tau_k$ is the quantile. $\beta(\tau_k)$ expresses the parameters to be estimated at the quantile $\tau_k$; $Q_y$ indicates the dependent variable; x is the vector of independent variables.

We can estimate Equation (4) using the way proposed by Koenker [75], and the formula is expressed as follows:

$$\min_{(\alpha,\,\beta)} = \sum_{k=1}^{K} \sum_{t=1}^{T} \sum_{i=1}^{N} w_k \rho_{\tau_k}\left(y_{it} - \alpha_i - x_{it}^T \beta(\tau_k)\right) \tag{5}$$

where K, T, and N represent the number of quantiles, time, and cross sections, respectively; $w_k$ is the weight of the k-th quantile, and

$w_k = 1/k$; $\rho_{\tau_k}$ stand for the piecewise linear quantile loss function.

If the number of N is large relative to T, then a larger number of fixed effects exist. They will inflate the variability of other coefficient estimates. Setting the individual effect $\alpha_i$ as one of the regression parameters, it can be computed through the following equation:

$$\min_{(\alpha,\,\beta)} = \sum_{k=1}^{K} \sum_{t=1}^{T} \sum_{i=1}^{N} w_k \rho_{\tau_k}\left(y_{it} - \alpha_i - x_{it}^T \beta(\tau_k)\right) + \lambda \sum_{i}^{N} \alpha_i \tag{6}$$

where $\lambda$ is the tuning parameter and robustness of the parameter $\beta$ to be estimated. Following Damette and Delacote [76], we set $\lambda = 1$ in the current work.

Because the focus of our study is the contribution of CSR to the financial constraints of one firm, the conditional quantiles function for quantile τ can be defined as follows:

$$
\begin{aligned}
Q_\tau(FC_{it}|\alpha_i, \varepsilon_{it}, x_{it}) \\
= \alpha_\tau + \beta_{1\tau}CSR_{it-1} + \beta_{2\tau}SIZE_{it} \\
+ \beta_{3\tau}GROWTH_{it} + \beta_{4\tau}ROA_{it} + \beta_{5\tau}LEV_{it} + \beta_{6\tau}TEN_{it} \\
+ \beta_{7\tau}SOE_{it} + \beta_{8\tau}ABSEM_{it-1} + \beta_{9\tau}CSR * ABSEM_{it-1}\mu_t + \varepsilon_{it}
\end{aligned}
\tag{7}
$$

where $Q_\tau(FC_{it}|\alpha_i, \varepsilon_{it}, x_{it})$ is the τ-th quantile regression function on FC; $\alpha\tau$ are a set of constant terms at each quantile τ; $\beta\tau$ is coefficient estimates corresponding to each quantile τ. We assign the values 0.05, 0.25, 0.5, 0.75, and 0.95 to the quartiles of τ. The detailed represents of the variables are provided in Appendix A.

Firms with better social responsibility are more likely to attract investors and have a larger investor support than those with bad publicity. Hence, the former will be relieved from financial constraints. For Hypothesis 1, as a result of using cash flows to represent the level of financial constraints: the greater the cash flows, the smaller the financial constraint will be. We expect the main effect of CSR to be positive. A positive coefficient will prove our hypothesis that the firms with good CSR disclosures could be relieved from financial constraints. We also predict that when the firm faces stronger financial constraints, that is, τ is in a smaller quantile, the absolute value of the coefficient of $\beta_{1\tau}$ will be larger than other quantiles. ABSEM is measured in such a way that larger values correspond to higher levels of financial opacity. Hence, we predict the main influence of ANSEM to be negative, and the coefficients on CSR*ABSEM are predicted to be positive.

We also use the Two-Stage Least Squares (2SLS) estimation method and two-stage Generalized Method of Moments (2step-GMM) to control for endogeneity for the robustness check. Furthermore, we re-estimate the empirical model with alternative definitions of FC and CSR to identity whether the results are sensitive to the presence of these measurement issues or not.

## 4. Empirical Results

### 4.1. Descriptive Statistics

The descriptive statistics of full sample are showed in Panel A of Table 1. The variables' name, the sample sizes, and the mean values are shown in the first three columns of each group. The standard deviation, the minimum values, and the maximum values of each variable are shown in the last three columns. The average of FC and CSR is 0.0562 and 40.3 respectively for the 2170 observations. The standard deviation of CSR is relatively high, which represents a large cross-sectional variation in the CSR engagement of firms ranging from 21.31 to 79.67. In other words, the overall level of CSR of publicly listed companies in China is still not high, and large differences exist among enterprises. The mean value of the variable SOE is 0.643. Hence, about 64% of the sample companies are SOEs. The average value of ABSEM is 0.0477 (min = 0.00054, max = 0.232), which shows that the sample company has a certain degree of information asymmetry. In addition, the average ROA is 5.87%, and the standard deviation is 5.277% (min = −11.91, max = 23.33). Thus, not all of the current disclosure of social responsibility reports in China come from high-performing companies. Some even suffered serious losses. Moreover, the distribution level of the remaining control variables is also basically reasonable, which is basically similar to the statistical results of existing research.

**Table 1.** Descriptive statistics. The descriptive statistics of the full sample is shown in Panel A of Table 1. The variables' names, the sample sizes, and the mean values are provided in the first three columns of each group. The standard deviation, minimum values, and maximum values of each variable are shown in the last three columns. The summary statistics and the significance of the mean and median difference between SOEs and non-SOEs are stated in Panel B of Table 1.

| Panel A: Full Sample Descriptive Statistics | | | | | |
|---|---|---|---|---|---|
| **Variables** | **N** | **Mean** | **Std.** | **Min** | **Max** |
| **FC** | 2170 | 0.0562 | 0.0747 | −0.181 | 0.257 |
| **CSR** | 2170 | 40.3 | 12.06 | 21.31 | 79.67 |
| **ABSEM** | 2170 | 0.0477 | 0.0451 | 0.00054 | 0.232 |
| **SIZE** | 2170 | 23.4 | 1.504 | 20.52 | 27.36 |
| **GROWTH** | 2170 | 9.212 | 23.51 | −46.89 | 98.99 |
| **SOE** | 2170 | 0.643 | 0.479 | 0 | 1 |
| **ROA** | 2170 | 5.87 | 5.277 | −11.91 | 23.33 |
| **LEV** | 2170 | 50.82 | 20.2 | 7.344 | 89.98 |
| **TEN** | 2170 | 58.69 | 16.45 | 21.93 | 93.82 |
| **INST** | 2170 | 52.58 | 21.34 | 1.629 | 91.91 |
| **FUND** | 2170 | 4.221 | 5.186 | 0 | 26.99 |
| **QFII** | 2170 | 0.182 | 0.567 | 0 | 3.454 |
| **SECUR** | 2170 | 0.147 | 0.472 | 0 | 2.814 |
| **SA** | 2170 | −3.764 | 0.31 | −4.362 | −2.458 |

| Panel B: Descriptive Statistics of the Subsample | | | | | | |
|---|---|---|---|---|---|---|
| **Variables** | **Mean Difference Test** | | | **Median Difference Test** | | |
| | **Non-SOEs** | **SOEs** | **MeanDiff** | **Non-SOEs** | **SOEs** | **Chi2** |
| **FC** | 0.06 | 0.05 | 0.01 * | 0.056 | 0.056 | 0.002 |
| **CSR** | 38.72 | 41.18 | −2.47 *** | 36.237 | 37.726 | 6.987 *** |
| **ABSEM** | 0.05 | 0.05 | 0 | 0.036 | 0.033 | 4.065 ** |
| **SIZE** | 22.83 | 23.72 | −0.89 *** | 22.703 | 23.664 | 158.488 ** |
| **GROWTH** | 12.48 | 7.4 | 5.08 *** | 8.674 | 5.342 | 14.502 *** |
| **SOE** | 0 | 1 | −1 | 0 | 1 | |
| **ROA** | 6.9 | 5.3 | 1.61 *** | 5.849 | 4.586 | 32.374 *** |
| **LEV** | 46.32 | 53.32 | −7.00 *** | 46.281 | 56.307 | 48.222 *** |
| **TEN** | 53.89 | 61.35 | −7.47 *** | 54.97 | 61.05 | 45.765 *** |
| **INST** | 42.45 | 58.21 | −15.76 *** | 42.918 | 60.458 | 181.851 ** |
| **FUND** | 5.07 | 3.75 | 1.32 *** | 3.022 | 2.168 | 13.827 *** |
| **QFII** | 0.17 | 0.19 | −0.01 | 0 | 0 | 10.610 *** |
| **SECUR** | 0.18 | 0.13 | 0.05 ** | 0 | 0 | 1.153 |
| **SA** | −3.85 | −3.71 | −0.14 *** | −3.839 | −3.775 | 31.362 *** |

*, **, and *** indicate statistical significance at the 10%, 5%, and 1% levels, respectively.

He et al. (2017) [77] confirmed the presence of heterogeneity in borrowing constraints between SOEs and non-SOEs. Hence, the characteristics of these two groups of firms are provided in Panel B of Table 1. The mean result and media difference test reveal significant differences in CSR disclosure,

company size, growth opportunity, fund investor, top 10 investors, institutional investor, and SA index between them.

The correlation matrix for our key variables is presented in Table 2. Comparing with prior studies [74,78], we find that the level of CSR disclosure is positively and significantly have relationship with cash flow (0.117). Thus, CSR firms are prefer to hold more cash than non-CSR firms. Hence, we can surmise that CSR disclosure could lessen FC. The relationship between SOE and CSR is significantly positive. Therefore, SOEs have a better CSR performance than non-SOEs.

**Table 2.** Correlation Analysis of the Key Variables.

| Variables | FC | CSR | ABSEM | SIZE | GROWTH | SOE | ROA | LEV | TEN |
|---|---|---|---|---|---|---|---|---|---|
| FC | 1 | | | | | | | | |
| CSR | 0.117 *** | 1 | | | | | | | |
| ABSEM | −0.201 *** | −0.129 *** | 1 | | | | | | |
| SIZE | −0.005 | 0.461 *** | −0.072 *** | 1 | | | | | |
| GROWTH | 0.044 ** | −0.019 | 0.037 * | 0.009 | 1 | | | | |
| SOE | −0.038 * | 0.098 *** | −0.01 | 0.284 *** | −0.104 *** | 1 | | | |
| ROA | 0.497 *** | 0.02 | −0.008 | −0.026 | 0.205 *** | −0.146 *** | 1 | | |
| LEV | −0.266 *** | 0.106 *** | 0.071 *** | 0.539 *** | 0.018 | 0.166 *** | −0.384 *** | 1 | |
| Ten | 0.137 *** | 0.308 *** | −0.021 | 0.408 *** | −0.036 * | 0.218 *** | 0.104 *** | 0.094 *** | 1 |

Note: *, **, and *** indicate statistical significance at the 10%, 5%, and 1% levels, respectively.

### 4.2. Conditional Mean Regression Results

We state the findings of Equation (3) in Table 3. We first use fixed-effects model of panel data to test the relationship between environmental information disclosure and financing constraints (According to the Hausman test, the panel data model with confirmed effect is more appropriate than the random effect model. For brevity, the test results are omitted). The results can be seen in columns (1)–(3) of Table 3. We only consider the most basic variables in the regression model in column (1), namely, cash flow (FC) and social responsibility information disclosure (CSR). To avoid the "pseudo-regression" problem caused by the time trend, we have controlled the year fixed effect. In addition, considering the differences in government governance levels and other regional characteristics in different regions, we have further controlled the province's fixed effects. Moreover, the test results may be highly correlated with the industry in which the company is located. Hence, we also control the fixed effects of the industry. As shown in Table 3, the coefficient of CSR is significantly positive. Its value is 0.000631, which indicates that corporate social responsibility information disclosure has significant financing constraint mitigation effect.

Because we only control the most basic variables, the above financing constraint mitigation effect may be brought about by other variables. Based on the above considerations, we control SIZE, GROWTH, SOE, ROA, and LEV, and so on. The CSR coefficient of column (2) is slightly reduced compared with column (1), but it remains significantly positive. We further include the hysteresis control variables in Table 3 (3), which include the lag phase 1 of SIZE, GROWTH, SOE, ROA, LEV, and TEN. The results still reflect significant financing constraint mitigation. The above results jointly support H1.

Columns (4) and (5) in Table 3 contain the test results of the impact of social responsibility information disclosure (CSR) on cash flows (FC) in different samples. The coefficient of CSR in non-state-owned enterprises is significantly positive. Its value is 0.000517. Thus, CSR can positively affect the company's cash flow, thereby alleviating financing constraints. By contrast, the coefficient of state-owned enterprises is only 0.00038, thereby showing that the influence of CSR is more pronounced in non-SOEs than in SOEs.

**Table 3.** Corporate social responsibility and financial constraints. The results of the fixed effect regression are shown in Table 3. The effect of CSR on the FC for the selected firms is shown in the table. The full sample results are listed in the first three columns. By contrast, the subsamples for non-SOEs and SOEs are shown in the last two columns, which test the differences in the influence of CSR on FC among the different property rights enterprises. t statistics are enclosed in parentheses.

| Dependent Variable= FCt. | Full Sample (1) | Full Sample (2) | Full Sample (3) | SOEs (4) | Non-SOEs (5) |
|---|---|---|---|---|---|
| **CSR** | 0.000631 *** | 0.000435 *** | 0.000359 ** | 0.000380 ** | 0.000517 ** |
| | (4.95) | (3.42) | (2.40) | (2.05) | (2.03) |
| **SIZE** | | −0.0000280 | −0.0352 ** | 0.00319 | −0.00814 *** |
| | | (−0.02) | (−2.45) | (1.57) | (−2.75) |
| **GROWTH** | | −0.0000231 | 0.0000670 | 0.00015 | −0.000253 * |
| | | (−0.28) | (0.68) | (1.43) | (−1.86) |
| **SOE** | | 0.00366 | 0.00341 | | |
| | | (1.14) | (0.97) | | |
| **ROA** | | 0.00663 *** | 0.00648 *** | 0.0054 *** | 0.00835 *** |
| | | (19.68) | (12.80) | (11.68) | (15.03) |
| **LEV** | | −0.000139 | −0.000194 | −0.000182 | −0.0000180 |
| | | (−1.36) | (0.66) | (−1.43) | (−0.09) |
| **TEN** | | 0.000344 *** | 0.000954 *** | 0.000182 | 0.000269 |
| | | (3.47) | (2.74) | (1.32) | (1.59) |
| **CONSTANT** | 0.0508 *** | 0.00673 | 0.0402 | −0.0407 | 0.139 * |
| | (4.23) | (0.21) | (1.13) | (−0.99) | (1.77) |
| **Lagged control variable** | N | N | Y | N | N |
| **Year and province effects** | Y | Y | Y | Y | Y |
| **Industry effects** | Y | Y | Y | Y | Y |
| *N* | 2170 | 2170 | 1736 | 1395 | 775 |
| *R2 adjusted* | 0.1619 | 0.3591 | 0.3593 | 0.3571 | 0.4406 |

*, **, and *** indicate statistical significance at the 10%, 5%, and 1% levels, respectively.

## 4.3. Quantile Regression Results

As studied earlier, the OLS model described the average relationship between CSR disclosures and FC based on the conditional mean of cash flows. The impact of CSR disclosures may change throughout the distribution of cash flows. Thus, we use QR in Equation (7) to infer information on the co-movement between CSR disclosures and FC in specific cash conditions, as reported in the upper half of Table 4. The QR coefficient estimates are plotted in Figure 1, with 95% confidence interval along with the OLS estimates. As for quantile coefficient estimates, we plot the nine-quantile regression estimates for $\tau = 0.1$ to 0.9 as the solid green curve with 95% confidence interval (shaded area). In particular, the results indicate that changing the conditional distribution, the effect of CSR disclosures changes on cash flows varies in size.

Table 4 and Figure 1 suggests that social responsibility performance have a positive effect on cash flows at four quantile levels (5th, 25th, 50th, and 75th), except for at the 95th. The results are tally with these stated in the mean regression models, i.e., the companies with good CSR performance can get cash more easily than those with poor CSR, which could alleviate financing constraints. However, one difference could be ignored. When the firm's cash flows at the highest quantile levels, the positive impact of CSR on cash flow is very small and insignificant. The effect of CSR disclosures at different quantile numbers is different. The coefficient is 0.000615 at 5th percentile. It decreases to 0.000437 at the 25th percentile and further reduces to 0.000387 and 0.000319 at the 50th and 75th percentile of firm cash flows, respectively. The effect of CSR on firm's cash flows is larger at low percentiles than at high percentiles. These results support for H2.

**Table 4.** The effcet of CSR on FC: quantile regression models.

| | Q5 | Q25 | Q50 | Q75 | Q95 |
|---|---|---|---|---|---|
| **Dependent Variable = FC** | (1) | (2) | (3) | (4) | (5) |
| **CSR** | 0.000615 ** | 0.000437 *** | 0.000387 *** | 0.000319 ** | 0.00000500 |
| | (1.87) | (2.29) | (2.85) | (2.17) | (0.02) |
| **SIZE** | 0.00890 *** | 0.00123 | −0.000154 | −0.00256 * | −0.00440 |
| | (2.57) | (0.61) | (−0.11) | (−1.65) | (−1.46) |
| **GROWTH** | −0.000151 | −0.000117 | 0.0000140 | 0.0000892 | 0.000410 ** |
| | (−1.03) | (−1.38) | (0.23) | (1.36) | (3.23) |
| **SOE** | 0.00868 | 0.00444 | 0.00428 | 0.000759 | 0.0126 |
| | (1.10) | (0.97) | (1.31) | (0.21) | (1.83) |
| **ROA** | 0.00290 *** | 0.00627 *** | 0.00729 *** | 0.00729 *** | 0.00624 *** |
| | (3.82) | (14.24) | (23.28) | (21.46) | (9.48) |
| **LEV** | −0.000966 *** | −0.0000360 | 0.0000904 | 0.000174 | 0.000196 |
| | (−3.90) | (−0.25) | (0.88) | (1.57) | (0.91) |
| **TEN** | 0.000334 | 0.000172 | 0.000165 | 0.000429 *** | 0.000596 ** |
| | (1.40) | (1.24) | (1.68) | (4.02) | (2.88) |
| **CONSTANT** | −0.274 *** | −0.0517 | 0.0141 | 0.0855 *** | 0.136 ** |
| | (−3.74) | (−1.22) | (0.47) | (2.61) | (2.14) |
| **Year and province effects** | Y | Y | Y | Y | Y |
| **Industry effects** | Y | Y | Y | Y | Y |
| *N* | 2170 | 2170 | 2170 | 2170 | 2170 |
| **Pseudo R2** | 0.2958 | 0.2024 | 0.2242 | 0.2681 | 0.3056 |

Note: *, **, and *** indicate statistical significance at the 10%, 5%, and 1% levels, respectively.

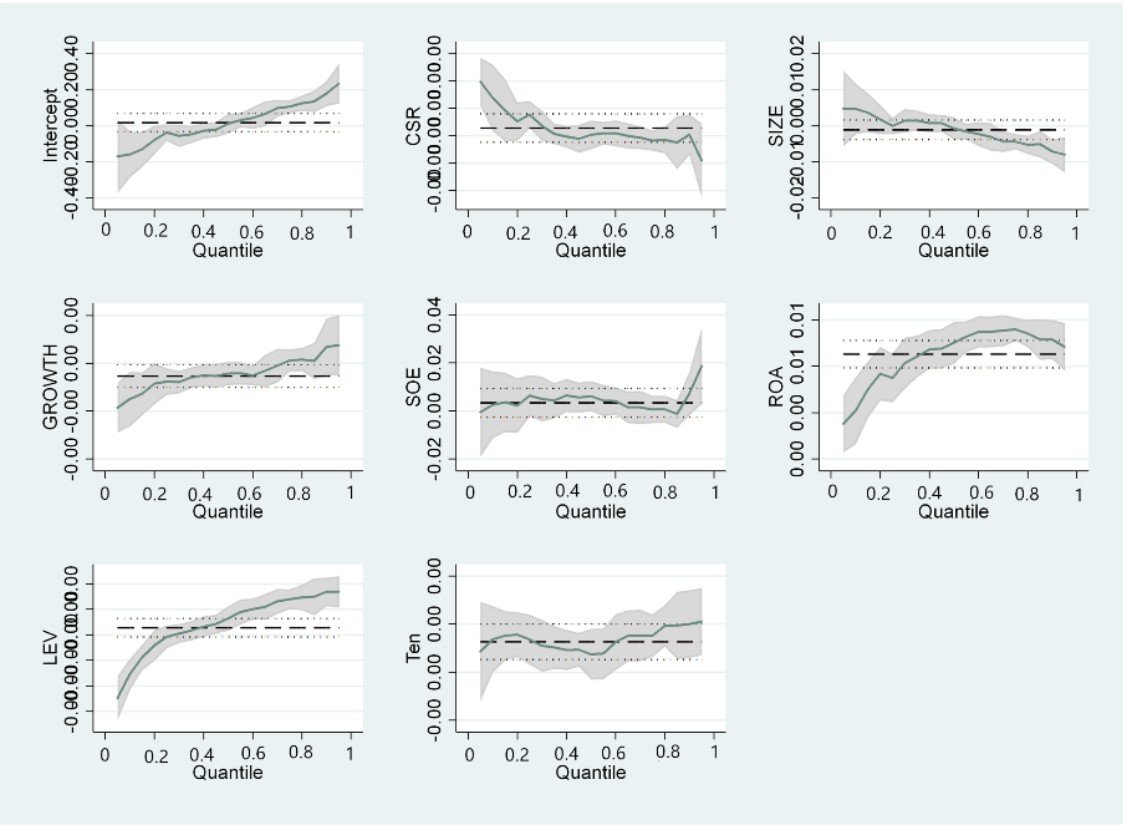

**Figure 1.** Quantile regression coefficient estimates for CSR scores change (linear model). QR estimates for τ = 0.1–0.9 are provided by the solid green curve with 95% confidence intervals (shaded area) for the influence of CSR disclosure on cash flows.

As regards the key control variables, ROA shows a significant positive impact. It passes the 1% significance level. Thus, ROA is a very important determinant of cash flow, which can effectively alleviate financing constraints. We found no significant impact from GROWTH, apart from the 10%

quantile. The estimation coefficients of TEN at high quantiles (75th and 95th quantiles) satisfy the significant requirement roughly, whereas those at low and medium quantiles are not significant. LEV affects the cash flow only at low quantiles (5th quantile), with no significant effect at the other quantiles.

In the future, we aim to extend our studies to examine the connection between CSR and financial constraints. If social information disclosure has incremental information value and could alleviate financing constraints, then investors of enterprises with insufficient financial information disclosure need to rely on nonfinancial information for decision-making. Thus, the link between corporate social responsibility information disclosure and FC should be obvious. We further apply the quantile regression method to test the interaction effect of financial opacity on the relationship between CSR and FC by adding variable ABSEM and interaction variable CSR*ABSEM. The results are shown in Table 5. ABSEM loads significantly negatively at low quantiles (5th and 25th quantile). Thus, financial opacity decreases the cash flows of the firm, thereby increasing the degree of financing constraints. Furthermore, the coefficients on CSR * ABSEM are significantly positive at 5th and 25th quantiles. Combined with the positive coefficient on CSR (0.000545 and 0.000206), these results indicate that the positive connection between CSR and the cash flows is provided in firms that are financially opaque. Thus, a substitutive relationship exists between these two forms of disclosure, as predicted in H3 and H4.

**Table 5.** The effect of CSR and ABSEM on FC: quantile regression models.

| | OLS | Q5 | Q25 | Q50 | Q75 | Q95 |
|---|---|---|---|---|---|---|
| **Dependent Variable= FC** | **(1)** | **(1)** | **(2)** | **(3)** | **(4)** | **(5)** |
| **CSR** | 0.0003148 * | 0.000545 * | 0.000206 * | 0.000433 ** | 0.000299 * | –0.00000788 |
| | (1.78) | (1.48) | (1.86) | (2.20) | (1.47) | (–0.02) |
| **ABSEM** | –0.2925248 *** | –0.433 * | –0.461 ** | –0.0974 | –0.0562 | –0.0694 |
| | (–2.68) | (–1.91) | (–3.15) | (–0.81) | (–0.45) | (–0.29) |
| **CSR*ABSEM** | 0.0014761 * | 0.000981 * | 0.00227 * | 0.00333 | 0.000362 | 0.00115 |
| | (1.53) | (1.77) | (1.61) | (0.88) | (0.11) | (0.19) |
| **SIZE** | –0.0009476 | 0.00808 ** | 0.00144 | –0.000117 | –0.00265 * | –0.00499 |
| | (–0.67) | (2.76) | (0.76) | (–0.08) | (–1.64) | (–1.60) |
| **GROWTH** | –0.0000251 | –0.000196 | –0.0000764 | –0.00000634 | 0.0000863 | 0.000399 *** |
| | (–0.42) | (–1.59) | (–0.96) | (–0.10) | (1.27) | (3.04) |
| **SOE** | 0.0037218 | –0.000640 | 0.00370 | 0.00551 | 0.00151 | 0.0123 |
| | (1.16) | (–0.10) | (0.86) | (1.55) | (0.41) | (1.73) |
| **ROA** | 0.0067233 *** | 0.00376 *** | 0.00648 *** | 0.00735 *** | 0.00724 *** | 0.00621 *** |
| | (21.78) | (5.86) | (15.64) | (21.46) | (20.45) | (9.08) |
| **LEV** | –0.000073 | –0.000852 *** | –0.0000399 | 0.0000822 | 0.000162 | 0.000185 |
| | (–0.72) | (–4.07) | (–0.29) | (0.74) | (1.41) | (0.83) |
| **TEN** | 0.0003695 *** | 0.000264 | 0.000142 | 0.000229 * | 0.000435 *** | 0.000655 *** |
| | (3.82) | (1.31) | (1.09) | (2.13) | (3.92) | (3.06) |
| **CONSTANT** | 0.03749 | –0.209 *** | –0.0396 | 0.00861 | 0.0902 ** | 0.155 * |
| | (1.24) | (–3.33) | (–0.98) | (0.26) | (2.61) | (2.32) |
| **Year and province effects** | Y | Y | Y | Y | Y | Y |
| **Industry effects** | Y | Y | Y | Y | Y | Y |
| *N* | 2170 | 2170 | 2170 | 2170 | 2170 | 2170 |
| **Pseudo R2** | 0.3776 | 0.3299 | 0.2307 | 0.2324 | 0.2684 | 0.3058 |

Note: *, **, and *** indicate statistical significance at the 10%, 5%, and 1% levels, respectively.

## 4.4. Endogeneity Tests and Robustness Checks

The relationship between CSR and FC may be determined by third-party factors, such as corporate governance. Therefore, endogeneity problems may arise due to the omission of important explanatory variables. Referring to the research Dhaliwal et al. (2011) [52], we introduce institutional investors share the total (INST) in Column (1) of Table 6 on the basis of controlling the fixed effect of the company and year, in Column (2) further control QFII shareholding ratio (QFII), Column (3) re-control the shareholding ratio indicators, namely, the fund shareholding ratio (FUND) and the securities institution's shareholding ratio (SECUR). We show the results with these additional variables in Columns (1)–(3) of Table 6. As shown in the Table, the FC coefficient is still significantly positive.

It reflects the mitigation effect of the company's social responsibility information disclosure on FC, which is consistent with our previous states.

**Table 6.** Robustness test by controlling additional variables and adopting two-step GMM methods.

| Dependent Variable = FC | FE | FE | FE | two-step GMM |
|:---:|:---:|:---:|:---:|:---:|
| | **(1)** | **(2)** | **(3)** | **(4)** |
| **CSR** | 0.000430 *** | 0.000402 *** | 0.000401 *** | 0.00493 * |
| | (3.36) | (3.18) | (3.16) | (1.95) |
| **SIZE** | –0.000308 | –0.000960 | –0.000948 | –0.0197 * |
| | (–0.20) | (–0.63) | (–0.62) | (–1.85) |
| **GROWTH** | –0.0000163 | –0.0000192 | –0.0000194 | –0.0000918 |
| | (–0.20) | (–0.23) | (–0.23) | (–1.23) |
| **SOE** | 0.00216 | 0.00172 | 0.00168 | 0.00781 |
| | (0.64) | (0.51) | (0.50) | (1.48) |
| **ROA** | 0.00651 *** | 0.00627 *** | 0.00627 *** | 0.00711 *** |
| | (18.50) | (17.79) | (17.52) | (12.07) |
| **LEV** | –0.000160 | –0.000120 | –0.000121 | 0.000212 |
| | (–1.54) | (–1.16) | (–1.17) | (0.62) |
| **TEN** | 0.000211 * | 0.000242 | 0.000242 * | –0.000167 |
| | (1.67) | (1.94) | (1.90) | (–0.63) |
| **INST** | 0.000186 * | 0.000162 | 0.000162 | 0.0000569 |
| | (1.69) | (1.48) | (1.47) | (0.46) |
| **QFII** | | 0.0129 *** | 0.0129 *** | 0.0103 *** |
| | | (5.37) | (5.36) | (3.35) |
| **FUND** | | | –0.00000450 | –0.000594 |
| | | | (–0.01) | (–1.68) |
| **SECUR** | | | –0.000880 | 0.0000770 |
| | | | (–0.33) | (0.02) |
| **L.FC** | | | | 0.0112 |
| | | | | (0.49) |
| **CONSTANT** | 0.0145 | 0.0262 | 0.0261 | 0.268 |
| | (0.46) | (0.84) | (0.84) | (1.90) |
| *N* | 2170 | 2170 | 2170 | 2165 |

Note: *, **, and *** indicate statistical significance at the 10%, 5%, and 1% levels, respectively.

"Good firms," which are likely to public a CSR report, also burden fewer financial constraints than "bad firms." Although we have controlled a number of variables in the regression equations, but the endogeneity is still possible between CSR and FC. We re-examine the link between CSR disclosures and FC using two-step GMM estimation methods. The results are shown in Columns (4) in Table 6. The significantly positive coefficients on CSR are consistent with our predictions, which are similar to those reported above. Again, our main results remain unchanged.

Further, we use a variable, lag1 period of CSR (called L.CSR), as our instrumental variable in a two-stage least square (2SLS) regression. We present the instrumental variable results in Table 7. Columns (1) and (3) show that the L.CSR variables are significant to predict CSR. We then use the predicted CSR variable (called L.CSR) to replace the CSR variable. Columns (2) and (4) indicate that the coefficients of the instrumented CSR variable (L.CSR) are positive and significant at the 1% level. The findings are consistent with those in Tables 3 and 6. The related test statistics (weak instrumental variable F-statistics) suggest that the instrument variables are appropriate.

**Table 7.** CSR and FC: Two-stage least square (2SLS) estimation.

|  | CSR | CF | CSR | CF |
|---|---|---|---|---|
|  | (1) | (2) | (3) | (4) |
| CSR |  | 0.0006615 *** |  | 0.0006488 *** |
|  |  | (4.07) |  | (4.00) |
| SIZE | 0.4050164 *** | −0.0028195 * | 0.4240119 *** | −0.0033431 ** |
|  | (4.55) | (−1.82) | (4.71) | (−2.14) |
| GROWTH | −0.0142518 *** | −0.0001434 ** | −0.0162147 *** | −0.0001135 * |
|  | (−3.65) | (−2.15) | (−4.12) | (−1.69) |
| SOE | −0.0514747 | 0.0049477 | 0.0108743 | 0.0029743 |
|  | (−0.26) | (1.45) | (0.05) | (0.83) |
| ROA | 0.0150807 | 0.0066348 *** | 0.0061658 | 0.0066314 *** |
|  | (0.76) | (19.62) | (0.30) | (18.74) |
| LEV | −0.0101149 * | −0.0002709 *** | −0.0109306 * | −0.0002616 ** |
|  | (−1.67) | (−2.61) | (−1.81) | (−2.52) |
| TEN | 0.0014205 | 0.0004273 *** | 0.0046476 | 0.0003114 ** |
|  | (0.23) | (3.97) | (0.57) | (2.24) |
| INST |  |  | −0.0022624 | 0.0001346 |
|  |  |  | (−0.34) | (1.18) |
| FUND |  |  | 0.0736687 *** | −0.0009234 *** |
|  |  |  | (3.60) | (−2.64) |
| QFII |  |  | −0.2239577 * | 0.00479 ** |
|  |  |  | (−1.80) | (2.25) |
| SECUR |  |  | 0.0281483 | −0.0014497 |
|  |  |  | (0.17) | (−0.52) |
| L.CSR | 0.8957517 *** |  | 0.895938 *** |  |
|  | (104.97) |  | (105.19) |  |
| CONSTANT | −3.455484 ** | 0.0448665 | −4.155013 ** | 0.0607895 ** |
|  | (−2.05) | (1.55) | (−2.42) | (2.06) |
| *Weak IV F Statistics* | 11,018.9 |  | 11,065.8 |  |
| N | 1736 | 1736 | 1736 | 1736 |
| $R^2$ | 0.8997 | 0.2702 | 0.9005 | 0.276 |
| F | 2213.41 |  | 1419.16 |  |

*, **, and *** indicate statistical significance at the 10%, 5%, and 1% levels, respectively. All variables are as defined in the Appendix A.

We have obtained the financing constraint mitigation effect of social responsibility information disclosure. However, the level of CSR is obtained by manual scoring. The measurement result may be affected by the subjective judgment of the coder, thereby affecting the robustness of the estimation result. As such, we classify the level of CSR from high to low into five groups (levels). We not only distinguish the effects of different disclosure levels, but also reduce the errors that may be caused by content analysis. Therefore, in Equation (3), we use the social information disclosure level variable (quart_CSR) instead of CSR for regression. The findings are shown in Table 8. We still discover similar results in Table 4. The role of quart_CSR on cash flow is significantly positive. It could alleviate financing constraints and the financial opacity positively affects the relationship between them. Therefore, after considering the measurement error of CSR, the conclusions of our paper are still relatively stable.

We also use alternative measures for FC to testify the robustness of our finding. The SA index is substituted by the cash flow. The robustness results taking various measures as replacement of FC which are presented in Table 9. A similar pattern is observed with our empirical results. The coefficient of CSR is significantly positive. Thus, the increase in firms' CSR disclosure could raise the SA index, thereby reducing firms' financial constraints.

**Table 8.** The Results of CSR Disclosure Measured as Quart_CSR.

| Dependent Variable = FC | (1) | (2) | (3) | (4) | (5) |
|---|---|---|---|---|---|
| Quart_CSR | 0.00281 | 0.00239 | 0.00186 | 0.000858 | −0.00541 [*] |
| | (0.82) | (1.28) | (1.19) | (0.52) | (−1.73) |
| ABSEM | −0.563 [**] | −0.462 [***] | −0.206 [*] | −0.228 [*] | −0.378 |
| | (−2.20) | (−3.33) | (−1.77) | (−1.86) | (−1.62) |
| CSR*ABSEM | 0.00406 | 0.00219 | −0.000462 | 0.00461 | 0.00998 [*] |
| | (0.63) | (0.62) | (−0.16) | (1.48) | (1.69) |
| SIZE | 0.00639 [*] | 0.000925 | 0.000488 | −0.00427 [***] | −0.00579 [*] |
| | (1.90) | (0.50) | (0.32) | (−2.64) | (−1.88) |
| GROWTH | −0.000163 | −0.0000359 | 0.0000110 | 0.0000770 | 0.000257 [*] |
| | (−1.12) | (−0.45) | (0.17) | (1.10) | (1.93) |
| SOE | −0.00576 | −0.00304 | −0.000133 | 0.00214 | 0.00858 |
| | (−0.72) | (−0.70) | (−0.04) | (0.56) | (1.17) |
| ROA | 0.00327 [***] | 0.00602 [***] | 0.00663 [***] | 0.00724 [***] | 0.00617 [***] |
| | (4.17) | (14.12) | (18.59) | (19.23) | (8.62) |
| LEV | −0.000815 [***] | −0.0000633 | 0.0000626 | 0.000305 [***] | 0.000423 [*] |
| | (−3.31) | (−0.47) | (0.56) | (2.58) | (1.88) |
| TEN | 0.0000270 | −0.0000307 | 0.000120 | 0.000430 [**] | 0.000769 [***] |
| | (0.09) | (−0.20) | (0.91) | (3.09) | (2.92) |
| INST | 0.000459 [*] | 0.000300 [**] | 0.000164 | 0.00000795 | −0.000183 |
| | (1.98) | (2.38) | (1.56) | (0.07) | (−0.87) |
| QFII | 0.0138 [**] | 0.0149 [***] | 0.0115 [***] | 0.0118 [***] | 0.0156 [***] |
| | (2.33) | (4.63) | (4.28) | (4.15) | (2.88) |
| FUND | −0.000947 | −0.000386 | 0.000146 | 0.000388 | −0.000386 |
| | (−1.35) | (−1.01) | (0.46) | (1.15) | (−0.60) |
| SECUR | −0.00370 | −0.00277 | −0.0000926 | −0.00109 | −0.00445 |
| | (−0.55) | (−0.75) | (−0.03) | (−0.34) | (−0.72) |
| CONSTANT | −0.157 [**] | −0.0212 | 0.0148 | 0.124 [***] | 0.179 [***] |
| | (−2.12) | (−0.53) | (0.44) | (3.48) | (2.64) |
| N | 2170 | 2170 | 2170 | 2170 | 2170 |

Note: *, **, and *** indicate statistical significance at the 10%, 5%, and 1% levels, respectively.

**Table 9.** Results of financial constraints measured as SA index.

| Dependent Variable = SA | (1) | (2) | (3) | (4) | (5) |
|---|---|---|---|---|---|
| CSR | 0.00232 [**] | 0.00360 [***] | 0.00558 [***] | 0.00614 [***] | 0.00517 [***] |
| | (2.54) | (3.65) | (7.62) | (8.13) | (5.37) |
| ABSEM | 0.677 | 1.081 [*] | 1.085 [*] | 0.842 [*] | 1.126 [*] |
| | (1.20) | (1.78) | (2.41) | (1.81) | (1.90) |
| CSR*ABSEM | −0.0229 | −0.0395 [**] | −0.0386 [***] | −0.0264 [**] | −0.0273 [*] |
| | (−1.60) | (−2.57) | (−3.38) | (−2.24) | (−1.82) |
| SIZE | 0.0847 [***] | 0.102 [***] | 0.0938 [***] | 0.0957 [***] | 0.101 [***] |
| | (11.54) | (12.87) | (15.95) | (15.78) | (13.13) |
| GROWTH | 0.000155 | 0.000198 | −0.0000282 | 0.000435 | 0.000379 |
| | (0.50) | (0.59) | (−0.11) | (1.69) | (1.16) |
| SOE | 0.0162 | 0.0247 | 0.0131 | 0.00915 | −0.00153 |
| | (0.95) | (1.34) | (0.96) | (0.65) | (−0.09) |
| ROA | −0.00320 [*] | −0.00250 | −0.00183 | −0.00438 [**] | −0.00661 [***] |
| | (−1.92) | (−1.39) | (−1.37) | (−3.18) | (−3.77) |
| LEV | −0.000896 [*] | −0.00216 [***] | −0.00150 [***] | −0.00263 [***] | −0.00270 [***] |
| | (−1.71) | (−3.81) | (−3.58) | (−6.06) | (−4.90) |
| TEN | 0.00212 [***] | 0.00362 [***] | 0.00476 [***] | 0.00539 [***] | 0.00599 [***] |
| | (3.45) | (5.45) | (9.64) | (10.59) | (9.26) |
| INST | −0.000716 | −0.00134 [**] | −0.00181 [***] | −0.00176 [***] | −0.000963 |
| | (−1.45) | (−2.51) | (−4.59) | (−4.33) | (−1.86) |
| QFII | −0.0119 | −0.0490 [***] | −0.00638 | −0.00604 | −0.0132 [*] |
| | (−0.95) | (−3.62) | (−0.63) | (−0.58) | (−1.00) |
| FUND | −0.000282 | 0.000191 | 0.00105 | −0.000874 | −0.00282 |
| | (−0.19) | (0.12) | (0.88) | (−0.71) | (−1.80) |
| SECUR | 0.00428 | −0.00232 | −0.00234 | −0.0136 | −0.00932 |
| | (0.30) | (−0.15) | (−0.20) | (−1.14) | (−0.62) |
| CONSTANT | −6.269 [***] | −6.349 [***] | −6.175 [***] | −6.048 [***] | −5.955 [***] |
| | (−39.87) | (−37.43) | (−49.00) | (−46.51) | (−36.01) |
| N | 2170 | 2170 | 2170 | 2170 | 2170 |

Note: *, **, and *** indicate statistical significance at the 10%, 5%, and 1% levels, respectively.

## 5. Conclusions

In our study, we tested the impact of social responsibility information disclosure on the financial constraints and the role of financial transparency to CSR engagement. We adopt panel fixed effect model and panel quantile regression method to examine company annual data over the period of 2013–2017. Our sample included 434 companies with social responsibility disclosure scores in RKS. we found that disclosure on social report influences negatively financial constraints. This influence is more obvious in firms with high levels of financial opaqueness than those with low financial opaqueness. Our study contributes to the research on the economic consequences of nonfinancial CSR disclosure [79]. In addition, the effect of CSR disclosure on FC is greater when the company faces stronger financial constraints. These results are robust to control numerous potentially confounding factors and adopt 2SLS and two-step GMM estimation methods.

We use fixed-effects model of panel data to test the relationship between CSR information disclosure and financing constraints. When we only consider the most basic variables in the regression model, namely, cash flow and social responsibility information disclosure (CSR). the coefficient of CSR is significantly positive. Its value is 0.000631, which indicates that corporate social responsibility information disclosure has significant financing constraint mitigation effect. We further include the hysteresis control variables, which include the lag phase 1 of SIZE, GROWTH, SOE, ROA, LEV, and TEN. The results still reflect significant financing constraint mitigation. The above results jointly support H1. Further, the coefficient of CSR in non-state-owned enterprises is significantly positive. Its value is 0.000517. Thus, CSR can positively affect the company's cash flow, thereby alleviating financing constraints. By contrast, the coefficient of state-owned enterprises is only 0.00038, thereby showing that the influence of CSR is more pronounced in non-SOEs than in SOEs.

Then, we use QR to infer information on the co-movement between CSR disclosures and FC in specific cash conditions, the results suggest that social responsibility performance have a positive effect on cash flows at four quantile levels (5th, 25th, 50th, and 75th), except for at the 95th. These are consistent with these stated in the mean regression models, i.e., the companies with good CSR performance can get cash more easily than those with poor CSR, which could alleviate financing constraints. However, one difference could be ignored. When the firm's cash flows at the highest quantile levels, the positive impact of CSR on cash flow is very small and insignificant. The effect of CSR disclosures at different quantile points is different. The coefficient is 0.000615 at 5th percentile. It decreases to 0.000437 at the 25th percentile and further reduces to 0.000387 and 0.000319 at the 50th and 75th percentile of firm cash flows, respectively. The effect of CSR on firm's cash flows is larger at low percentiles than at high percentiles. These results support H2. We further apply the quantile regression method to test the interaction effect of financial opacity on the relationship between CSR and FC by adding variable ABSEM and interaction variable CSR*ABSEM. ABSEM loads significantly negatively at low quantiles (5th and 25th quantile). Thus, financial opacity decreases the cash flows of the firm, thereby increasing the degree of financing constraints. Furthermore, the coefficients on CSR * ABSEM are significantly positive at the 5th and 25th quantiles. Combined with the positive coefficient on CSR (0.000545 and 0.000206), these results indicate that the positive connection between CSR and the cash flows is provided in firms that are financially opaque.

From the methodological perspective, we adopt the quantile regression method to test the differential effect of social performance on FC at different cash flows distributions. The calculation of the quantile regression estimator is based on the absolute residual minimization of an asymmetric form. Compared with ordinary least squares OLS regression, the fractional regression can more comprehensively describe the whole picture of the conditional distribution of explained variables and analyze how the explained variables affect the explained variables. Quantile regression does not require strong assumptions for error terms, so for non-normal distributions, quantile regression coefficient estimators are more robust. Our paper also augments prior empirical studies utilizing the QR method to further highlight merits and applicability of this method in economics and finance study, which we hope may stimulate additional research along this line.

Our study also has significant practical implications. Results confirm that social performance can affect the behavior of external fund providers. If the level of CSR disclosure can be closely related to the performance of the company, then the disclosure behavior will be proactive. This outcome will help stakeholders conduct positive interactions around social responsibility information. At present, China's requirements for CSR information disclosure are still not detailed enough. The company has a greater choice of the specific content of the disclosure than the government. This situation will not make the disclosure of CSR to be trustworthy, comparable, and comprehensive, thereby limiting the role of information disclosure. The regulatory authorities may consider further clarifying the standards for CSR, improving its usefulness, and ultimately using economic means to guide the company to improve social responsibility consciously.

Authors of future works should study if the content of the social responsibility information report is objective. They must verify the authenticity and integrity of the CSR disclosure, thereby revealing the intention of the discloser to a certain extent. In the future, machine learning and text mining methods can be adapted to evaluate the subjectivity of social responsibility and financial reports through which we can obtain accurate influences of CSR and financial disclosure on firms.

**Author Contributions:** Conceptualization, C.L. and B.D.; Data curation, B.D., L.G. and H.C.; Formal analysis, N.L.; Investigation, C.L.; Methodology, N.L. and Q.G.; Resources, L.G.; Software, N.L. and H.C.; Supervision, Q.G.; Writing–original draft, N.L.; Writing–review & editing, C.L. and B.D.

**Funding:** This research was supported by "the Fundamental Research Funds for the Central Universities" (Grant No. 2019BSCX16).

**Acknowledgments:** The authors are grateful for the financial support provided by "the Fundamental Research Funds for the Central Universities" (Grant No. 2019BSCX16).

**Conflicts of Interest:** The authors declare no conflict of interest.

**Appendix A**

**Table A1.** Variable definitions.

| Variables | Symbols | Definition |
|---|---|---|
| **Dependent variables** | **FC** | Net cash flow from operating activities /Net fixed value in the previous period |
| | **SA** | $-0.737*\text{Size} + 0.043*\text{Size}^2 - 0.04Age$ |
| **Independent variables** | **CSR** | Social responsibility disclosure score comes from the Hexun website |
| | **quart_CSR** | Five average groups of CSR |
| | **ABSEM** | Used to measure the transparency of accounting information at the company level by modifying the Jones model to obtain the absolute value of the manipulated accruals |
| **Control variables** | **SIZE** | The natural logarithm of total assets |
| | **GROWTH** | Growth rate equals operating income minus lagged operating income scaled by lagged operating income |
| | **SOE** | State-owned enterprises, state = 1; otherwise state = 0 |
| | **ROA** | Net income/total asset |
| | **LEV** | Total liabilities divided by total assets. |
| | **TEN** | Total share ratio of the top ten shareholders |
| | **INST** | Total share ratio of institution investors |
| | **FUND** | Total share ratio of fund companies |
| | **QFII** | Total share ratio of QFII |
| | **SECUR** | Total share ratio of securities companies |

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
