# Peer review of "Corporate Social Responsibility and Financial Performance: A Quantile Regression Approach"

_sustainability, doi:10.3390/su11133717_

Round 1
Reviewer 1 Report
This paper is about the impact of CSR disclosure on financial constraints. In my opinion, the issue is interesting but the paper is too unbalanced on quantitative analysis, missing the conceptual framework within the data should be interpreted. For this reason, I think it do not contribute enough to the advancement of the literature review.
First, the title does not reflect the real content of the paper: it is too vague. In the abstract I expect to see the theoretical end/or practical impacts of the study. The abstract must capture the attention of the reader. The last lines (20-21) do not reflect what you are going to say in the following part of the article.
Introduction. L.29: why don’t you go further in the analysis? Year 2013 is a time too outdated. L.51: explain acronym the first time you use it (I sense FC is Financial Constraints, but you should say it)
Literature review. This part is very weak in my opinion. L. 114-116: You recall some famous theory about corporate governance, but the authors you cite are not really the ‘fathers’ of these theories. See, for example, Jensen and Meckling for principal-agent theory and Freeman for stakeholder theory. Which theory do you embrace and why? L.117-125: you refer to CSR performance. First, it is not clear what do you mean for CSR performance…in this paper a part about this issue is missing. CSR is not a matter of course and you should explain the link between CSR and sustainability, considering that they are not synonymous. In the title you talk about ‘financial sustainable development’ but what do you mean for sustainable finance? Furthermore, when you refer to CSR performance, you should explain how to measure that performance (there are different opinions about this matter) L. 126-133: there are also more studies that prove a positive relations CSR information disclosure and capital markets. L. 156 and L. 163: what is ‘high quality information disclosure’? How could you measure the quality of non financial information? For example, there are international recognized standards, as GRI, that ensure the quality of ESG information. L.161: not always CSR disclosure serves as a substitutive information source for financial disclosure (e.g. in the integrated reporting financial and socio-environmental issues are read together.
The quantitative analysis is well done.
Finally, I suggest to review the editing of references in the text.
Author Response
At first, we appreciate editors’ and reviewers’ valuable comments and suggestions which help us improve the paper significantly. Please find our responses to your comments in the attached file. Thank you very much.

Reviewer 2 Report
In general terms, the paper under review appears good. However, some issues should be improved prior to definitely accepting it.
To be precise:
- At least a slight review by an English native or expert would be advisable, as there are some punctual grammar or typing mistakes along the text.
- The authors refer in different parts of their paper to the "negative association" between CSR disclosure and financial constraints in a company (in the abstract for the first time, and then several times along the paper). This is a semantic issue as well as a questions on preferences, but sometimes may be something like "CSR disclosure influences negatively financial constraints" could be an alternative (just a suggestion). This way, the existence of CSR disclosure influence would be presented as more "active" in terms of reducing financial constraints.
- The first statement in the introductory sector (page 1, lines 26-27) is clearly questionable. Arguing that "rapid economic development has resulted in an increasing awareness of social and environmental problems" is not a generalized statement at all. Maybe something like "consciousness on the consequences of rapid economic development..." would be a better alternative.
- The second sentence (page 1, lines 27-29) refer to "rankings" including figures for 2008, 2009 and 2013, but no source is mentioned or concrete information on the ranking is provide (and it should be). Moreover, only an increase in absolute figures is stated, but no mention to the relative importance or representativeness of such figures is provided (e.g. percentage on total publicly Chinese listed companies). And, additionally, information on most recent years (2013-2019) would be welcome.
- In page 1, line 38, there is a quotation of a contribution by Cormier et al. (2009), which is not included in the final list of bibliographical references. It should have been identified as [1] and included as the first reference in the final list. There is a similar situation in other places of the text. What is more, please note that, according to Sustainability format guidelines, bibliographical references should be identified with Arabic and not Roman figures in numbers, i.e., as [1] instead of [i]. A similar situation (quoted bibliographical references without identifying numbers and then not included in the final list) appears, e.g., in page 5 (lines 202 -twice- and 209-210), in page 7 (line 303), in page 8 (line 314), in page 9 (line 347), or in page 14 (line 457).
- In page 1, line 44, we can see the quotation of the contribution by Clarkson et al. (2008), which is identified as [i] and then included as the first one in the final list of bibliographical references. On the one hand, it should be identified as [2] and included as the second reference in this final list (see the previous comment on the reference by Cornier et al., 2009). On the other hand, and much more relevant, in case of a bibliographical quotation, and accordingly to Sustainability format guidelines, only the identifying number will appear and the mention to the authors' surnames and the year of their publication will not. So, in this case, it should be "...and is rewarded by society [2]". This change (eliminating authors' names and years) should be done in most cases all along the paper.
- In page 2, line 51, we can see the acronym FC for the first time in the text. As in case of other acronyms (e.g. CSR), the complete word should appear together to the acronym when used for the first time. To be precise, a potential reader would not find this "clarification" until page 5. Similarly, the meaning if the acronym "RKS" in line 76 is not provided.
- In page 2, line 59, the way in which the sentence is redacted provides a chance to include the surname of the authors and the publishing year of the quoted reference (ok), but then the identifying number should appear together to the complete mention of the quoted contribution instead of at the end of the sentence, that is, "Cheung et al. (2018) [5] presented...".
- The contribution by Francis et al. (2005) is identified as [7] and [30] in the final list, i.e., it appears twice on there. then the second one it appears, should be removed from the list. Additionally, remember on joining the quotation and the identifying number as indicated in the previous comment: "Francis et al. (2005) [7] found that...". A review of the whole text is required, in order to: a) eliminate names and years while only identifying numbers (in Arabic figures) remain; or b) indicate the identifying numbers just together to the name and year of the quotation (instead of locating the numbers at the end of the sentence). For example, in page 3, line 112, we can see a case (maybe the only one) when the quotation is adequate according to Sustainability format guidelines: "Richardson et al. (1999) [13] found that...".
- In page 3, line 119, there are two identifying numbers ([20] and [21]) for three bibliographical references (Dhaliwal, et al. 2011; El Ghoul, et al. 2011; Lai, et al. 2017). On the one hand, as previously mentioned and repeated, only identifying numbers should be provided in this case (according to the redaction of the sentence). On the other hand, three identifying numbers should have been provided. Similarly, only one identifying number is provided for two quoted contributions in le 130 (unless this number [27] is linked to the contribution by Platonova et al. in 2008, and the identifying number for the previous contribution by Frankle in 1980 is missed). The same situation appears, e.g., in page 4 (line 159).
- The quotation of the contribution by Dittmar and Mahrt-Smith in 2007 (page 6, line 244, as well as in page 8, line 314) should be identified as [35] (as it is included in the final list, but the identifying number does not appear in the text). Similarly, the contribution by Damette and Delacote (2012) in line 265 seems to be the number [36], the contribution by El Ghoul et al. (2011) in line 274 seems to be the number [4], or the contributions by Clarkson et al. (2008) and Dhaliwal et al. (2011) in page 12 (line 406) seem to be numbers [1] and [2] (or [2] and [3], if previously including -as required- the contribution by Cormier et al. in 2009 as number [1]).
- Finally, note than only a few of the references included in the final list are dated in the most recent period 2017-2019 or 2015-2019. Thus, review and consideration of more updated bibliographical references would increase the value of the paper (of course, this being a suggestion).
Author Response
Response to Reviewer Nana Liu, Chuanzhe Liu , Quan Guo , Bowen Da, Linna Guan and Huiying Chen At first, we appreciate editors’ and reviewers’ valuable comments and suggestions which help us improve the paper significantly. Response to Reviewer 2 Comments Point 1: In general terms, the paper under review appears good. However, some issues should be improved prior to definitely accepting it. To be precise: - At least a slight review by an English native or expert would be advisable, as there are some punctual grammar or typing mistakes along the text. Response1: Thanks for the suggestion. After careful reading, we found that the article does have some punctual grammar and typing mistakes. According to this point, we asked the help to the professional English editor, and revised the all incorrect words and sentences, or some expressions are not professional. We believe that there is no problem with the English language of this manuscript. Point 2: The authors refer in different parts of their paper to the "negative association" between CSR disclosure and financial constraints in a company (in the abstract for the first time, and then several times along the paper). This is a semantic issue as well as a questions on preferences, but sometimes may be something like "CSR disclosure influences negatively financial constraints" could be an alternative (just a suggestion). This way, the existence of CSR disclosure influence would be presented as more "active" in terms of reducing financial constraints. Response 2: Thanks for the suggestion. I greatly agree with your valuable advices. Based on your suggestion, we have made the following changes: “Abstract: A prominent claim within the literature is that corporate social responsibility disclosured firms are fundamentally more resilient to financial shocks, relative to firms that take no corporate social responsibility action. …We find that CSR disclosure influences negatively financial constraints. The quantile regression results also indicate that this influences would more obvious when a company faces stronger financial constraints. Further, CSR disclosure influences negatively financial constraints in financially opaque firms,and the effect of financial opaque on the relationship strengthens when the company faces great financial constraints...” We have rewritten the hypothesis as follows: “ We predict that CSR disclosure will help investors fully understand the future value of enterprises and alleviate adverse selection. …Our predictions are as followings: Hypothesis 1:. CSR information disclosure influences negatively financial constraints. Hypothesis 2: The negative influences is more obvious when company faces stronger financial constraints. ” “Impression management theory by Elsbach and Sutton (1992) believes that a company with bad financial performance may conduct corporate impression management by disclosing nonfinancial information… Hypothesis 3: financial transparency negative influences the relationship between CSR disclosure and FC. Hypothesis 4: The negative effect of financial transparency on the relationship between CSR disclosure and FC strengthens when the company faces stronger financial constraints.” In the conclusion part, we also made changes. “In our study, we tested the impact of social responsibility information disclosure …we found that disclosure on social report influences negatively financial constraints. This influence was more obvious in firms with high levels of financial opaqueness than those with low financial opaqueness. Our study contributes to the research on the economic consequences of nonfinancial CSR disclosure…” Point 3: - The first statement in the introductory sector (page 1, lines 26-27) is clearly questionable. Arguing that "rapid economic development has resulted in an increasing awareness of social and environmental problems" is not a generalized statement at all. Maybe something like "consciousness on the consequences of rapid economic development..." would be a better alternative. Response3: Thanks for your advices, and we agree with your opinion. We have changed the first statement in the introductory sector” Rapid economic development has resulted in an increasing awareness of social and environmental problems.” as “There has been increasing consciousness on the consequences of rapid economic development such as social and environmental problems.“ Point 4:- The second sentence (page 1, lines 27-29) refer to "rankings" including figures for 2008, 2009 and 2013, but no source is mentioned or concrete information on the ranking is provide (and it should be). Moreover, only an increase in absolute figures is stated, but no mention to the relative importance or representativeness of such figures is provided (e.g. percentage on total publicly Chinese listed companies). And, additionally, information on most recent years (2013-2019) would be welcome. Response4: Thank you for your advice, and I think your advice is necessary. We have added the percentage on total publicly Chinese listed companies and the figures in recent years. “According to the results of Rankins corporate social responsibility Ratings(RKS), 290 publicly listed Chinese companies disclosed corporate social responsibility (CSR) reports in 2008 , which increased to 371 in 2009 and to 851 in 2018. The " Environmental, Social and Governance of Chinese Listed Companies (ESG) Blue Book (2018)" published by the Joint Research Group of the Chinese Academy of Social Sciences and the Responsible Cloud Research Institute revealed the fact that among the 1892 A-share main board listed companies, there were 673 social responsibility reports in 2017, accounting for only 35.57%, and 145 reports were below 10 pages.” Point 5: - In page 1, line 38, there is a quotation of a contribution by Cormier et al. (2009), which is not included in the final list of bibliographical references. It should have been identified as [1] and included as the first reference in the final list. There is a similar situation in other places of the text. What is more, please note that, according to Sustainability format guidelines, bibliographical references should be identified with Arabic and not Roman figures in numbers, i.e., as [1] instead of [i]. A similar situation (quoted bibliographical references without identifying numbers and then not included in the final list) appears, e.g., in page 5 (lines 202 -twice- and 209-210), in page 7 (line 303), in page 8 (line 314), in page 9 (line 347), or in page 14 (line 457). Response5: Thanks for the suggestion. We are sorry for my careless, and we noticed the problems, then we have revised the format of bibliographical references according to Sustainability format guidelines and have added the reference literatures in the final list of bibliographical references. Point 6: - In page 1, line 44, we can see the quotation of the contribution by Clarkson et al. (2008), which is identified as [i] and then included as the first one in the final list of bibliographical references. On the one hand, it should be identified as [2] and included as the second reference in this final list (see the previous comment on the reference by Cornier et al., 2009). On the other hand, and much more relevant, in case of a bibliographical quotation, and accordingly to Sustainability format guidelines, only the identifying number will appear and the mention to the authors' surnames and the year of their publication will not. So, in this case, it should be "...and is rewarded by society [2]". This change (eliminating authors' names and years) should be done in most cases all along the paper. Response6: Thanks for the suggestion. Based on your comments, we have added a list of bibliographical references, and changed the Roman figures to Arabic numbers according to the Sustainable Format Guide. And in the text, in most cases, the author's name and year are eliminated, such as the lines 188-189 and lines163-164. Similarly, we have made corrections in the rest in this paper. Point 7: - In page 2, line 51, we can see the acronym FC for the first time in the text. As in case of other acronyms (e.g. CSR), the complete word should appear together to the acronym when used for the first time. To be precise, a potential reader would not find this "clarification" until page 5. Similarly, the meaning if the acronym "RKS" in line 76 is not provided. Response7: Thank you for your advice, and we think your advice is necessary. The acronym FC for the first time in the text is appeared in the abstract, we have added the complete word of FC. “Abstract: A prominent claim within the literature is that …To test this, we examine the impact of corporate social responsibility information disclosure(CSR) on financial constraints (FC). Our sample is composed of ….” Similarly, we have added the full meaning of RKS in line 42. “There has been increasing consciousness…According to the results of Rankins corporate social responsibility Ratings(RKS) ,290 publicly listed Chinese listed companies disclosed corporate social responsibility (CSR) reports in 2008 , which increased to 371 in 2009 and to 851 in 2018….” Point 8: - In page 2, line 59, the way in which the sentence is redacted provides a chance to include the surname of the authors and the publishing year of the quoted reference (ok), but then the identifying number should appear together to the complete mention of the quoted contribution instead of at the end of the sentence, that is, "Cheung et al. (2018) [5] presented...". Response8: Thanks for the suggestion. We have made corrections according to your revised comments, such as in lines 170, “Our research contributes to extant study…Francis, et al. [16] found that the level of transparency as reflected in the firm’s financial reports is negatively linked with the cost of equity capital..…. the interactive features of the two different kinds of disclosure ways.” Similarly, we have made corrections in the rest. “Corporate social responsibility, as both a concept and practice, has become popular in the researches. Backman (1975)[17], Davis (1960)[18] and Manne (1972)[19] defined the meaning of CSR by considering social responsibility as the opposite of economic responsibility. McGuiire (1963)[20] divided the responsibilities of enterprises into economic, social and legal responsibility. Social responsibility mainly means that enterprises should pay attention to politics, social welfare, education, employee benefits and other social interests. Preston (1975)[21] first analyzed CSR based on a process perspective and proposed a “corporate social responsibility matrix”. Some scholars began to systematically integrate relevant theoretical results, and gradually formed the concept of "corporate social performance". Carroll (1979)[22]constructed a three-dimensional model of corporate social performance. Wartick and Cochran (1985)[23] inherited Carroll's three-dimensional model. They believe that identifying social issues is important, but more important is how to manage social issues. Wood (1991)[24] mainly revised the principle dimensions of the Carroll model.…” Point 9: - The contribution by Francis et al. (2005) is identified as [7] and [30] in the final list, i.e., it appears twice on there. then the second one it appears, should be removed from the list. Additionally, remember on joining the quotation and the identifying number as indicated in the previous comment: "Francis et al. (2005) [7] found that...". A review of the whole text is required, in order to: a) eliminate names and years while only identifying numbers (in Arabic figures) remain; or b) indicate the identifying numbers just together to the name and year of the quotation (instead of locating the numbers at the end of the sentence). For example, in page 3, line 112, we can see a case (maybe the only one) when the quotation is adequate according to Sustainability format guidelines: "Richardson et al. (1999) [13] found that...". Response9: Thanks for the suggestion. We have examined the whole text and corrected the reference format in accordance with the Sustainability Format Guide,and marked in red in the revised manuscript. Point 10:- In page 3, line 119, there are two identifying numbers ([20] and [21]) for three bibliographical references (Dhaliwal, et al. 2011; El Ghoul, et al. 2011; Lai, et al. 2017). On the one hand, as previously mentioned and repeated, only identifying numbers should be provided in this case (according to the redaction of the sentence). On the other hand, three identifying numbers should have been provided. Similarly, only one identifying number is provided for two quoted contributions in le 130 (unless this number [27] is linked to the contribution by Platonova et al. in 2008, and the identifying number for the previous contribution by Frankle in 1980 is missed). The same situation appears, e.g., in page 4 (line 159). Response 10 : Thanks for the suggestion. We added missing references to the revised draft,and provided the corresponding number in the revised manuscript. “Regarding the possible links between CSR and economic reward, most of studies found that CSR could significantly relieve corporate FC[52, 53].” In lines 253-259, “By contrast, CSR information disclosure may put the company at a disadvantage in the market competition. Anderson and Frankle [30] concluded that excessive CSR information disclosure could also negatively impact the whole capital market. Platonova et al. (2018)[58] use OLS estimator approach to analyze the CSR related data. Their findings indicated a significant negative link between CSR and financial performance of commercial banks in Vietnam. In addition, based on the analysis of information asymmetry theory, an interactive U-type nonlinear relationship may exist between them, and some studies argue that there are no concern about CSR and firm performance[60].” In lines 287-211, “Numerous studies found a negative relationship between financial transparency level and the cost of equity in U.S.[61]. Francis et al. (2005)[16]and Hail and Leuz (2006)[62] extended this research scale to international settings. Dhaliwal et al. (2012)[63] believed that the connection between analyst forecast error and CSR disclosure is not positive in companies and countries owned great financial opacity.” Similarly, we have made corrections in the rest and marked in red in the revised manuscript. Point 11: - The quotation of the contribution by Dittmar and Mahrt-Smith in 2007 (page 6, line 244, as well as in page 8, line 314) should be identified as [35] (as it is included in the final list, but the identifying number does not appear in the text). Similarly, the contribution by Damette and Delacote (2012) in line 265 seems to be the number [36], the contribution by El Ghoul et al. (2011) in line 274 seems to be the number [4], or the contributions by Clarkson et al. (2008) and Dhaliwal et al. (2011) in page 12 (line 406) seem to be numbers [1] and [2] (or [2] and [3], if previously including -as required- the contribution by Cormier et al. in 2009 as number [1]). Response 11 : Thanks for the suggestion. We are sorry for our careless, and we have revised the whole references for matching each other. Point 12: - Finally, note than only a few of the references included in the final list are dated in the most recent period 2017-2019 or 2015-2019. Thus, review and consideration of more updated bibliographical references would increase the value of the paper (of course, this being a suggestion). Response 12 : Thanks for your suggestion. we have added more updated bibliographical references in this paper. Such as: “Albuquerque R, Koskinen Y, Zhang C. Corporate social responsibility and firm risk: Theory and empirical evidence[J]. Management Science, 2018. Hamrouni A, Boussaada R, Ben Farhat Toumi N. Corporate social responsibility disclosure and debt financing[J]. Journal of Applied Accounting Research, 2019. Francis B, Harper P, Kumar S. The effects of institutional corporate social responsibility on bank loans[J]. Business & Society, 2018, 57(7): 1407-1439. Cui J, Jo H, Na H. Does corporate social responsibility affect information asymmetry?[J]. Journal of Business Ethics, 2018, 148(3): 549-572.” On behalf of co-authors, we thank you very much for giving us an opportunity to revise our manuscript, we appreciate you very much for your positive and constructive comments and suggestions on our manuscript. We are also very grateful to you for your reference. In the future research, we will be more rigorous and careful.
Reviewer 3 Report
The aim of the reviewed manuscript is to examine the impact of CSR information disclosure on financial constraints. I think this paper has the potential to contribute to Sustainability. However, I have some deep concerns and I think the article needs to undergo a lot of revisions:
(1) The abstract of the article needs to be rewritten. The limitations of the research should be included.
(2) Keywords should be improved. Please see other similar papers published in Sustainability.
(3) The section of introduction should be improved. The way the article is positioned could be improved.
(4) Some of the claims of the article are not well justified. See for example the first sentece of the article: “Rapid economic development has resulted in an increasing awareness of social and 26 environmental problems.“ See also the following claim: “ The 38 disclosure of CSR information is a form of “defensive disclosure,” which is intended mainly to 39 defend a company’s own behavior.” Please improve these points.
(5) The author/s refer/s to a gap in the literature. More specifically, they ascertain the following: ‘However, few researches have focused on the relationship between CSR disclosure and FC. FC refer to the obstacles that prevent enterprises from investing in the expected projects.’ Nevertheless, the gap they/he/she try to address it’s not clear at all, specially, if the mentioned review and update flaws are considered. Please improve these points substantively.
(6) Some of the references included in the section of introduction and in the literature review are not updated and/or are not authoritative. Author/s should improve this issue, considering also previous remarks.
(7) Again, I would like to highlight that the theoretical and literature review needs to be better clarified so that the authors can clearly articulate the originality of the current study. In other words, the authors should improve the embedding of the reviewed work in the existing scholarly literature. This should be improved.
(8) The way the hypotheses are justified is not clear at all. Please improve them in the light of the scholarly specialized literature (e.g. Hypothesis 1b, Hypothesis 2b).
(9) In my view the author/s should improve the section of conclusions of the article substantively. This should be made in an improved theoretical framework and considering the updated literature review (please see previous remarks). This is a very relevant point in this Reviewer’s perspective.
(10) Likewise, author/s should improve the section of limitations of their article. This is in my view one of the most relevant flaws of the article. One of the main limitation is related to the endogeneity problems related to the quantitative analysis carried out by the author/s of the study. This issue should be analyzed in depth.
.
To put it briefly, I think this is an interesting work with potential. The information provided in the article is interesting and the authors have gathered good data but the work has to be improved substantively.
Author Response

(The authors gave the same response as above.)

Round 2
Reviewer 1 Report
I appreciate very much the efforts made by authors to improve their paper. I think the current version is more rigorous. I finally suggest to include in the references some articles published in the same journal you are writing for (in this case Sustainability) to highlight you are in line with the spirit and the aims of the journal. I wish the authors the best for their future researches.
Author Response
Response to Reviewer 1 Comments Point 1: I appreciate very much the efforts made by authors to improve their paper. I think the current version is more rigorous. I finally suggest to include in the references some articles published in the same journal you are writing for (in this case Sustainability) to highlight you are in line with the spirit and the aims of the journal. I wish the authors the best for their future researches. Response1: Thanks for the suggestion. We greatly appreciate your feedback. According to your comments, We refer to some articles published on Sustainability that are closely related to this paper. In lines 252-255, we have added the sentences “Ok and Kim(2019)[60] found that firms with better corporate social responsibility (CSR) performance generally exhibit cheaper equity financing. Cupertino et al.(2019)[61]found that the environmental and social firm performance positively impacted corporate capital accumulation using a sample of US manufacturing firms from 2002 to 2017.” In lines 263-264, we added the sentences “Dai et al.(2019)[63]found that there is an inverted U-shaped nonlinear relationship between CSR information disclosure and stock price crash risk.” In line 698, we also refer to one article published in Sustainability. The references in the final list are “ Ok, Y.; Kim, J., Which Corporate Social Responsibility Performance Affects the Cost of Equity? Evidence from Korea. Sustainability 2019, 11, (10), 2947. Cupertino, S.; Consolandi, C.; Vercelli, A., Corporate Social Performance, Financialization, and Real Investment in US Manufacturing Firms. Sustainability 2019, 11, (7), 1836. Dai, J.; Lu, C.; Qi, J., Corporate Social Responsibility Disclosure and Stock Price Crash Risk: Evidence from China. Sustainability 2019, 11, (2), 448. Yao, S.; Liang, H., Analyst Following, Environmental Disclosure and Cost of Equity: Research Based on Industry Classification. Sustainability 2019, 11, (2), 300.” On behalf of co-authors, we thank you very much for giving us an opportunity to revise our manuscript, we appreciate you very much for your positive and constructive comments and suggestions on our manuscript. We are also very grateful to you for your reference. In the future research, we will be more rigorous and careful.
Reviewer 3 Report
All my suggestions and doubts have been resolved. The authors have done an excellent job in incorporating improvements to the article.
Author Response
Response to Reviewer 3 Comments Point : All my suggestions and doubts have been resolved. The authors have done an excellent job in incorporating improvements to the article. Response: On behalf of co-authors, we are very grateful to you for your reference. In the future research, we will be more rigorous and careful.